# The Shape of Success: A Scoping Review of Somatotype in Modern Elite Athletes Across Various Sports

**DOI:** 10.3390/sports13020038

**Published:** 2025-02-04

**Authors:** Ximena Martínez-Mireles, Edna Judith Nava-González, Manuel López-Cabanillas Lomelí, Debbie Samantha Puente-Hernández, Miriam Gutiérrez-López, José Omar Lagunes-Carrasco, Ricardo López-García, Erik Ramírez

**Affiliations:** 1Facultad de Salud Pública y Nutrición, Universidad Autónoma de Nuevo León, UANL, Av. Universidad S/N Ciudad Universitaria, San Nicolás de los Garza 66451, Nuevo León, Mexico; ximenamtznutricion@gmail.com (X.M.-M.); edna.navagn@uanl.edu.mx (E.J.N.-G.); manuel.lopezcabanillaslm@uanl.edu.mx (M.L.-C.L.); dpuenteh@uanl.edu.mx (D.S.P.-H.); myriam.gutierrezlop@uanl.edu.mx (M.G.-L.); 2Facultad de Organización Deportiva, Universidad Autónoma de Nuevo León, UANL, Av. Universidad S/N Ciudad Universitaria, San Nicolás de los Garza 66451, Nuevo León, Mexico; jose.lagunesca@uanl.edu.mx (J.O.L.-C.); ricardo.lopezgr@uanl.edu.mx (R.L.-G.)

**Keywords:** body composition, athletic performance, body physique, physical profiling, somatoplot, anthropometry

## Abstract

This scoping review aimed to determine the somatotype of modern elite athletes across various sports. The literature search followed the PRISMA guidelines for scoping reviews. Four databases were consulted, PubMed, Scopus, Web of Science, and Clarivate InCites, as well as platforms such as Google Scholar, Taylor & Francis Online, Books Ovid, CAB eBooks, MyiLibrary, and Core Collection. Eligibility criteria included articles published between 1995 and 2024 involving athletes at the elite level, such as junior, senior, Olympic, first-division professionals, world-class competitors, national competitors, black belts, master athletes, non-professional athletes with at least 10 years of experience, those ranked in the national or international top 10, and high-level collegiate athletes. A total of 66 studies were included. Modern male elite athletes predominantly exhibited an endomorphic mesomorphic somatotype, whereas female athletes were classified as central. The present study was the first to synthesize data from 66 articles, encompassing athletes of varying elite levels, both males and females, while providing detailed information on age, weight, height, BMI, somatotype and its classification, and complemented by somatoplot references. Future research is recommended to include a greater diversity of sports and a more significant number of elite female athletes to enhance the representativeness of elite athletes.

## 1. Introduction

Biotypology, the study of human types based on morphological, physiological, and psychological characteristics, has evolved significantly [1,2]. Within this discipline, body physique, which includes body size, structure, and composition, plays a crucial role in sports science and health by influencing athletic performance and overall physical fitness [3,4]. Sheldon et al. [5] introduced the concept of somatotype, categorizing body physiques into three components: endomorphy, mesomorphy, and ectomorphy. This classification emphasizes body shape and composition over size [6,7,8]. The somatotype method, later refined by Heath and Carter [9], is a widely recognized approach for assessing body physique. It involves anthropometric measurements such as skinfold thickness, girth measurements, and bone diameters to generate a three-number score representing the relative dominance of each somatotype component [10,11].

The somatotype has proven to be a determining factor in athletic performance, influencing strength, endurance, and sport-specific skills depending on the discipline [12]. For example, the mesomorphic and ectomorphic components significantly influence muscle torque and power in judokas [13]. Mesomorphism is positively associated with better sprint performance in activities like track and field, whereas endomorphism is negatively correlated with vertical jump scores. Additionally, mesomorphic and ectomorphic components enhance aerobic capacity, for example, in endurance sports. On the other hand, individuals with a central somatotype have been observed to achieve the highest values of anaerobic power. At the same time, mesomorphic ectomorphs exhibit the lowest performance levels in healthy people [14]. Over the past century, athletes have experienced significant increases in height and weight, surpassing the rates observed in the general population. For instance, National Basketball Association (NBA) players have shown notable increases in height, while National Football League (NFL) players have exhibited greater height and body mass. Additionally, body mass index (BMI) has increased in sprinters since 1900, whereas it has decreased in long-distance runners, reflecting a form of ’Darwinian’ selection pressure favoring optimal physical attributes for specific sports [15,16]. This trend, driven by self-selection based on anatomical traits, has resulted in more specialized physiques, showing more remarkable similarity among athletes within the same sport while displaying more pronounced differences between athletes from different sports [17]. Technological advancements, including artificial neural networks, 3D body scanning, and bioimpedance analysis, have improved the accuracy, efficiency, and accessibility of somatotype assessment [18,19,20], while the professionalization of sports and progress in training, nutrition, psychology, and ergogenic aids have distinguished modern athletes from those of the past, contributing to continuous record-breaking performances [21,22,23,24].

Male athletes tend to have a more mesomorphic somatotype, characterized by a muscular and robust build, while female athletes often exhibit a higher endomorphic component, indicating a more significant proportion of body fat and softer body composition [25,26]. Genetics and environment both play crucial roles in shaping somatotype development, with genetic factors predominantly influencing mesomorphic and ectomorphic components, while environmental factors have a more significant impact on the endomorphic component [27,28,29,30]. Mesomorphy is generally considered advantageous for strength and explosive power, while ectomorphy tends to offer benefits in endurance activities [31,32]. Understanding the somatotypes of modern elite athletes is therefore essential. For example, research indicates that body composition in athletes evolves due to training, nutrition, and sport-specific demands. A study on female collegiate soccer players reported increases in total mass, fat-free mass (FFM), fat mass, and body fat percentage (%BF) over four years [33]. Similarly, research on NCAA Division 1 female athletes found lean mass increased in volleyball and swimming athletes over three years, while basketball players experienced a rise in %BF [34]. This knowledge not only deepens our understanding of how physical characteristics influence sports performance but also offers practical applications. Addressing somatotypes in sports offers several advantages for injury prevention. It allows for the identification of injury-prone physique types, informs tailored training and positioning, aids in monitoring health risks, and enhances performance in contact sports [35,36,37,38].

By integrating somatotype analysis into athlete selection and training, sports professionals can reduce injury rates and improve overall athlete safety and performance. This knowledge can guide the selection and specialization of athletes based on their morphology, making it a valuable resource for coaches and talent scouts. Although the somatotypes of elite athletes in various sports have been studied, most of this research was conducted before 1995 or is limited to specific disciplines [12,39,40,41,42,43,44]. While these studies have provided valuable insights, focusing solely on specific disciplines or relying on older data may not adequately capture the evolving morphological characteristics influenced by advancements in training, nutrition, and technology in sports over the past decades. Furthermore, emerging sports and modern international competitions often present unique physical demands that require updated analyses. Consequently, the somatotypes of modern elite athletes remain insufficiently characterized. This gap in recent studies on diverse sports, age groups, countries, and elite levels underscores the need for a scoping review to establish updated reference patterns. These patterns could inform and guide future research as well as practical applications in sports, such as talent identification and the optimization of training and nutrition programs. Therefore, this scoping review aims to determine the somatotype of modern elite athletes across various sports: team sports, combat sports, speed and endurance sports, and individual sports. 

## 2. Materials and Methods

### 2.1. Protocol and Registration

A scoping review was conducted following the criteria outlined in the Preferred Reporting Items for Systematic Reviews and Meta-Analyses for Scoping Reviews (PRISMA-ScR) [45] and the methodological framework proposed by the Joanna Briggs Institute (JBI) [46] (Appendix A). Since this is a scoping review, it was not registered in PROSPERO.

### 2.2. Eligibility Criteria

Studies published between 1995 and 2024 were included to examine the evolution of elite athletes, whose morphological characteristics, developmental trajectories, and genetic and hormonal profiles have undergone substantial changes compared to earlier periods [15,47,48,49]. The year 1995 was chosen as the starting point for this review to reflect a more modern era in sports science. This period is characterized by significant advancements in training methodologies, nutritional strategies, and technological innovations, which have profoundly shaped the physical attributes and performance of elite athletes.

Articles in English and Spanish were selected. The sample included athletes aged 13 to 45 years, and one study reported an open age category. The rationale for including athletes from different age groups is to capture the full spectrum of an athlete’s career, from early development to peak performance. The included athletes represented a range of elite levels, including senior, Olympic, first-division professional athletes, world-class competitors, national competitors, black belts, and master athletes. Junior competitors and those ranked in the top 10 nationally or internationally were also included. Moreover, non-professional athletes with extensive competitive experience were eligible, specifically those with at least 10.0 years of competitive involvement, characterized as individuals who have consistently participated in competitive sporting activities, demonstrating sustained dedication and advanced performance in their discipline. High-level student-athletes and first-division National Collegiate Athletic Association athletes (NCAA) were also considered [50,51]. Body composition data were collected during initial, baseline, or pre-competition stages. To ensure a comprehensive understanding, all study types were included, without methodological restrictions, encompassing cross-sectional studies and case–control studies. This approach provides a diverse perspective on the topic, enhancing the thoroughness and comprehensiveness of the research.

Articles involving amateur, recreational, paralympic, preschool, and school-aged athletes (under 12 years old) were excluded, as well as those involving second and third-division athletes, individuals with less than 9.9 years of experience, and athletes participating in winter or extreme sports. Studies classified as gray literature were also excluded. Additionally, articles were excluded if they failed to specify the athlete’s sex, the sport practiced or if they combined data from recreational and elite-level participants.

### 2.3. Information Sources

The literature search was conducted using a combination of structured databases and platforms. Structured databases included PubMed, Scopus, Web of Science, and Clarivate InCites, which provide comprehensive and systematic indexing of peer-reviewed research. Platforms and tools utilized included Google Scholar, Taylor & Francis Online, Books Ovid, CAB eBooks, MyiLibrary, and Core Collection, which serve as access points or aggregators for scholarly content and resources. The study was conducted between April and September 2024. Only articles published between 1995 and 2024 were included in the analysis. Only peer-reviewed articles were included. The first four authors (X.M.-M, E.J.N.-G., M.L.-C.L., D.S.P.-H.) carried out the search strategy, which was discussed with the remaining authors. To ensure a comprehensive review, additional relevant references were identified through manual searches within the initially selected articles.

### 2.4. Search

The search strategy included the terms (“body composition” OR “anthropome*” OR “somatotype” OR “physique classification” OR “morphotype” OR “endomorphic” OR “mesomorphic” OR “ectomorphic”) AND (“elite athlete” OR “athlet*” OR “elite”) applied to titles and abstracts. Four authors (X.M.-M., E.J.N.-G., M.L.-C.L., D.S.P.-H.) conducted this search across all databases. Additionally, references from relevant articles in other reviews using the same search strategy were manually screened.

### 2.5. Selection of Sources of Evidence

The selection process for sources was conducted in three phases: removal of duplicates, screening of titles and abstracts to exclude non-relevant studies, and full-text review of potentially relevant studies. The initial screening was performed by the first four authors (X.M.-M.,E.J.N.-G., M.L.-C.L.,D.S.P.-H.) while the final eligibility assessment was carried out by all authors (X.M.-M., E.J.N.-G., M.L.-C.L., D.S.P.-H., M.G.-L., J.O.L.-C., R.L.-G.). Three meetings were held in August and September 2024, during which 93 articles were excluded.

### 2.6. Data Charting

Four authors (X.M.-M., E.J.N.-G., M.L.-C.L.D.S.P.-H.) conducted data extraction using a Microsoft^®^ Excel^®^ spreadsheet for Microsoft 365. Subsequently, the remaining authors (M.G.-L., J.O.L.-C., R.L.-G.)reviewed the accuracy of the data entry. Discrepancies during the review process were resolved through discussion among the first four authors, and consensus was reached after collaborative deliberation. If agreement could not be achieved, a third reviewer from the remaining authors was consulted to finalize the decision. We employed a systematic approach to ensure consistency in data charting by adhering to predefined criteria.

For the development of somatotype charts, the NutriSolver^®^ software version 1.0.0, Monterrey, N. L., Mexico [52] was used to determine the X and Y axes, employing the following equations:(1)X=ectomorphy component−endomorphy component(2)Y=2∗mesomorphy component−(endomorphy component+ectomorphy component)

### 2.7. Data Items

The data were recorded in a Microsoft^®^ Excel^®^ spreadsheet for Microsoft 365, including the first author’s last name, year of publication, article title, sport, study design, type of sport, athlete’s elite level, sample size, position, category, type of skinfold caliper, age, sex, body mass (kg), height (cm), body mass index (BMI), endomorphy, mesomorphy, ectomorphy, and somatotype classification. Sixty-six articles yielded 185 data on modern elite athletes. The Excel^®^ sheets were organized by sport and sex, and an additional consolidated sheet contained all the data.

### 2.8. Synthesis of Results

Tables were created to present the information collected on modern elite athletes. Data from 185 modern elite athletes were analyzed and plotted using NCSS 8 software (version 8.0.24, Kaysville, UT, USA) [53]. Sports were categorized into four groups: team sports, combat sports, speed and endurance sports, and individual sports. Additionally, weighted means were calculated for somatoplots by sports classification and in reference tables named “SomaRef” to obtain a representative average of the sample and ensure all articles were included in the analysis.

## 3. Results

### 3.1. Study Selection and Eligibility Assessment

A total of 155 articles were assessed for eligibility. Subsequently, 13 duplicate articles were removed from the database. The remaining studies (*n* = 142) were re-evaluated to confirm their eligibility. Ultimately, 80 articles were excluded, while 66 articles that met the inclusion criteria were incorporated into this scoping review, as illustrated in Figure 1.

### 3.2. Characteristics of Sources of Evidence

This scoping review included a total of 66 studies published between 1995 and 2024 [55,56,57,58,59,60,61,62,63,64,65,66,67,68,69,70,71,72,73,74,75,76,77,78,79,80,81,82,83,84,85,86,87,88,89,90,91,92,93,94,95,96,97,98,99,100,101,102,103,104,105,106,107,108,109,110,111,112,113,114,115,116,117,118,119], comprising a sample of 3757 modern elite athletes representing 43 sports disciplines, with a predominance of combat sports. The athletes’ ages ranged from 13.5 to 41.4 years. Overall, most athletes exhibited a predominant mesomorphy component.

The variables assessed included sport, elite level, reference, sample size, type of skinfold caliper, sex, age, body mass (kg), height (cm), BMI (kg/m^2^), endomorphy, mesomorphy, ectomorphy, and somatotype classification. Nine types of skinfold calipers were identified: Jamar^®^, Holtain^®^, Harpenden^®^, Lange^®^, Slim Guide^®^, John Bull^®^, Cescorf^®^, and GPM^®^. The most frequently used caliper in the studies was the Holtain^®^, followed by the Harpenden^®^ and the Jamar^®^.

All individual information on the athletes, including body composition characteristics, number of subjects, elite level, caliper used, somatotype components, and classification, is presented in Appendix A. Appendix A were developed to establish somatotype references, referred to as “SomaRef”. Additionally, Appendix A illustrate the somatotype charts of elite athletes based on sport classification.

Given the presence of various elite levels, judokas were selected as the focus group due to their representation across a diverse spectrum of elite levels, including international competitors, senior athletes, South Korean national-level athletes, master belts (1st to 5th dan), black belts, and distinct weight categories. This analysis revealed differences in somatotypes, as illustrated in Appendix A.

### 3.3. Results of Individual Source of Evidence

It was determined that most of male elite athletes were classified as endomorphic mesomorph (32.8%), followed by balanced mesomorph (25.2%), ectomorphic mesomorph (18.3%), mesomorph–ectomorph (6.9%), mesomorphic endomorph (3.8%), mesomorph–endomorph (3.8%), central (3.1%), balanced ectomorph (3.1%), mesomorphic ectomorph (2.3%), and endomorphic ectomorph (0.8%). For female elite athletes, the majority were classified as central (31.5%), followed by endomorphic mesomorph (22.2%), mesomorph–endomorph (20.4%), balanced ectomorph (5.6%), mesomorphic endomorph (5.6%), balanced mesomorph (5.6%), mesomorph–ectomorph (3.7%), endomorph–ectomorph (1.9%), balanced endomorph (1.9%), and ectomorphic mesomorph (1.9%). Variations in somatotypes according to sport type and sex are detailed in Appendix A, which serve as reference tables referred to as “SomaRef” in the Appendix A.

In team sports, male athletes predominantly exhibited an endomorphic mesomorph classification, with baseball and rugby players clustering within this somatotype. Specific variations were observed in volleyball, where setters and hitters tended toward balanced ectomorph (Appendix A). On the other hand, female athletes in team sports showed a predominant mesomorph–endomorph somatotype, with athletes such as water polo players, futsal players, basketball players, and soccer players exhibiting notable differences based on positions in futsal and volleyball. In volleyball, liberos were classified as endomorphic mesomorph (Appendix A).

In combat sports, male athletes predominantly exhibited an endomorphic mesomorph somatotype, which included ssireum wrestlers, Olympic wrestlers, jiu-jitsu athletes, pencak silat fighters, boxers, mixed martial arts athletes, karatekas, and judokas in the <90 kg, <100 kg, and >100 kg weight categories (Appendix A). In contrast, female athletes in combat sports were also primarily classified as endomorphic mesomorph, including karatekas, fencers, judokas, and judokas in the <48 kg, <70 kg, and <78 kg weight categories (Appendix A).

In speed and endurance sports, male athletes were predominantly grouped in the ectomorphic mesomorph classification. A notable variation was observed in a 42 km marathon runner who was classified as balanced ectomorph (Appendix A). Among female athletes, the predominant somatotype was central (Appendix A).

Finally, in individual sports, male athletes primarily exhibited an endomorphic mesomorph distribution, which included powerlifters in lightweight, middleweight, and heavyweight categories, surfers, and tennis players (Appendix A). Female athletes, on the other hand, were predominantly classified as central, including gymnasts, sprint paddle athletes, ten dance dancers, and Latin dance dancers (Appendix A).

### 3.4. Summary of Key Findings

The results indicated that the endomorphic mesomorph somatotype is predominant among male elite athletes, especially in combat sports (Figure 2). In contrast, female elite athletes were primarily classified as central, particularly in team sports (Figure 3).

In team sports, male athletes tend to exhibit an endomorphic mesomorph somatotype, while female athletes lean toward a mesomorph–endomorph classification. Variations are observed based on position in disciplines such as futsal and volleyball. In combat sports, both male and female athletes predominantly display an endomorphic mesomorph somatotype, though differences are noted across weight categories.

In speed and endurance sports, male athletes are concentrated mainly in the ectomorphic mesomorph classification, whereas female athletes tend toward a central classification. Lastly, in individual sports, the endomorphic mesomorph somatotype is common among males, particularly powerlifters, while females predominantly exhibit a central classification.

## 4. Discussion

### 4.1. Summary of Evidence

This scoping review included 66 studies covering 43 sports disciplines, published between 1995 and 2024. It was determined that the endomorphic mesomorph somatotype is predominant among male elite athletes (32.8%) (Figure 2), while the central somatotype is most common among female elite athletes (31.5%) (Figure 3). Previous studies on the somatotype of elite athletes have provided valuable insights; however, several limitations are evident. Some studies were published before 1995 [17,39,40,41,42], reflecting outdated training methodologies and athlete profiles. Others included only a limited range of sports, such as basketball and bodybuilding [43], or grouped athletes from different sports without sufficient differentiation, which reduced the specificity of their findings. Additionally, many studies lacked comprehensive data on key anthropometric variables such as age, weight, height, and BMI or failed to account for sex differences and elite competence levels [12,44]. In contrast, the present study is the first to synthesize data from 64 articles, encompassing athletes across various elite levels, both male and female, while providing detailed information on age, weight, height, BMI, somatotype classification, and references through somatoplots. Moreover, this study offers a classification by sport, enhancing the specificity and applicability of the findings by including distinctions based on position and category.

Athletes with an endomorphic mesomorphic somatotype exhibit high musculature and a higher percentage of body fat, which is advantageous in strength and power sports due to the balance of muscle mass and stability [82,120]. The central classification represents a balanced distribution of body components without a predominant characteristic [17]. A notable difference between the sexes was observed. Among the three components, males exhibited a higher mesomorphic component, while females tended toward endomorphy.

#### 4.1.1. Team Sports

In team sports, male athletes exhibited an endomorphic mesomorph somatotype (Appendix A), as seen in baseball players and rugby players. In baseball, body composition varies by position: pitchers tend to be taller and exhibit greater endomorphy, while first basemen and outfielders are typically heavier and more muscular, contributing to improved offensive performance. Infielders, particularly second basemen, have less muscle mass and lower body weight [67,121]. These findings underscore the importance of considering somatotypes in the selection and training of baseball players. In rugby, players predominantly exhibit an endomorphic mesomorph somatotype, reflecting the physical demands of the sport. Forwards are heavier and more muscular than backs, which enhances their absolute strength and power, crucial in high-intensity confrontations [122,123,124]. Age also impacts somatotype, with older players exhibiting higher levels of endomorphy [38,125,126,127,128].

In volleyball, somatotype variations were observed across positions. Opposites tend to have a balanced ectomorph somatotype, centers are mesomorph–ectomorph, and setters are endomorphic ectomorph. Centers and opposites are generally taller, heavier, and more robust, with superior jumping abilities. Setters and liberos, in contrast, are lighter and exhibit different strength profiles [129,130,131,132].

Among female athletes in team sports (Appendix A), the predominant somatotype was mesomorph–endomorph, observed in soccer players, handball players, basketball players, and futsal forwards. These athletes exhibit a combination of musculature and a higher percentage of body fat, consistent across different levels of competition and playing positions [133,134]. In futsal, notable differences were observed depending on the position: pivot wings, wingers, and goalkeepers displayed a mesomorphic endomorph somatotype, while pivots and forwards were classified as mesomorph–endomorphs. Goalkeepers and pivots tend to be more mesomorphic and have greater body mass. Over time, there has been a trend toward increased musculature and reduced body fat among futsal players. Additionally, reproductive factors influence body composition, as post-menarche athletes tend to exhibit higher body fat and lower ectomorphy [135]. Despite differences across sports, there is notable homogeneity within each discipline, particularly at the highest levels of competition [134]. This suggests that elite athletes tend to converge toward an optimal somatotype adapted to the specific demands of their sport.

#### 4.1.2. Combat Sports

Among male athletes, the predominant somatotype was endomorphic mesomorph (Appendix A), including ssireum wrestlers, Olympic wrestlers, jiu-jitsu athletes, pencak silat fighters, boxers, mixed martial arts athletes, karatekas, and judokas in the <90 kg, <100 kg, and >100 kg weight categories. These athletes exhibit key physical traits, such as high levels of strength, power, and anaerobic capacity [136]. Teenagers in combat sports often have more variability in body weight and height, with a higher prevalence of underweight individuals. Functional characteristics like cardiovascular reserve and muscle strength are influenced by body type and age [137,138]. Adults, on the other hand, show more stable body mass and composition, with performance being closely linked to fat-free mass and muscle tissue, including high strength, power, and anaerobic capacity [139,140]. While aerobic endurance and grip strength are essential, their significance varies depending on the sport. Speed plays a particularly critical role in striking disciplines. These attributes, previously studied in adult athletes, collectively contribute to high performance and success in physically demanding sports [74,141,142,143]. In addition, body physique varies by weight category.

Among elite female athletes, the endomorphic mesomorph somatotype also predominated (Appendix A), observed in karatekas, fencers, and judokas (<48 kg, <70 kg, and <78 kg categories). These athletes share characteristics such as a low body fat percentage, high muscle mass, and bone density, which are essential for their performance in sports requiring strength and endurance. However, significant variations in body dimensions and somatotypes exist across different combat sports, reflecting the specific physical demands of each discipline [144,145,146,147]. For instance, judokas tend to have a more robust build, with larger torsos and greater body circumferences. Taekwondo athletes are generally heavier and have higher conicity indices compared to their counterparts in judo and karate [148]. Overall, these athletes are lean, muscular, and possess high aerobic capacity, all critical factors for their success.

The observed differences in somatotypes among judokas (Appendix A) at various elite levels provide valuable insights into the physical demands and morphological adaptations required for success in judo. Elite-level judokas, including international competitors and senior athletes, often exhibit somatotypes that favor strength and power, such as a higher balanced mesomorphic profile, due to the intensive physical requirements of the sport. In contrast, South Korean national-level judokas and master belts (1st to 5th dan) displayed differences despite having the same weight category classifications, demonstrating variations not only by elite level but also by ethnicity. This reflects a combination of strength, endurance, and technical skill development, which varies across weight categories. Heavier weight categories may be associated with higher endomorphic components, whereas lighter categories favor ectomorphic or mesomorphic profiles. These findings underscore the importance of somatotype-specific training and nutrition programs to optimize performance at different competitive levels and weight classes. The inclusion of judokas with diverse elite levels in this analysis broadens our understanding of how body composition and somatotype influence performance outcomes in judo.

#### 4.1.3. Speed and Endurance Sports

In speed and endurance sports, the predominant somatotype among male athletes was endomorphic mesomorph (Appendix A), as observed in sprinters (100 m, 200 m, and 400 m), middle-distance runners (800 m, 1500 m), long-distance runners (3000 m, 5000 m, 10,000 m), and triathletes. Marathon runners (42 km) exhibit a high ectomorphic component, indicating a lean physique with minimal fat, advantageous for endurance [149,150]. High maximal oxygen uptake (VO_2_max) is crucial for marathon performance; In addition, they present better-running economy and higher lactate threshold velocities. Efficient use of energy sources, particularly a higher turnover rate in fat metabolism, is essential for sustaining long-duration efforts. This helps in delaying fatigue by conserving glycogen stores [151,152]. The interaction between muscles and tendons, especially in the lower leg, plays a significant role. A more considerable soleus muscle and thicker Achilles tendon contribute to better performance by enhancing force production and energy storage [153]. While benefiting from a lean build, racewalkers show greater variability in somatotype and tend to have higher body fat compared to middle- and long-distance runners [154]. Sprinters, on the other hand, display significant muscularity and lower body fat [155]. They also exhibit non-uniform hypertrophy, with significantly more significant hip and knee muscles compared to non-sprinters, which aids in generating greater force [156]. World-class sprinters optimize their stride length and frequency, maintain a smaller thigh angle at contact to shorten contact time, and exhibit a larger stride landing angle [157]. They also have a stiffer ankle joint and extend the knee throughout the stance phase, which aids in energy generation [158].

Among female athletes, the somatotypes of racewalkers and triathletes were classified as central (Appendix A). Other studies indicate that racewalkers exhibit a broader range of somatotypes, reflecting greater heterogeneity within this group [154,159,160]. Race walking involves increased electromyographic activity in the muscles of the trunk and lower limbs compared to fast walking. This heightened activity helps maintain the specific gait required for race walking [161]. Performance in race walking is associated with specific strength characteristics, such as dorsiflexor strength and knee flexor strength, which are critical for maintaining the required gait mechanics [162]. Understanding somatotype variations by discipline and gender is essential for optimizing performance and designing training programs tailored to the specific needs of speed and endurance sports.

#### 4.1.4. Individual Sports

Male athletes in individual sports predominantly exhibited an endomorphic mesomorph somatotype (Appendix A), including powerlifters in lightweight, middleweight, and heavyweight categories, tennis players, surfers, and pelotaris. Powerlifters typically demonstrate high mesomorphy, indicative of a muscular and robust physique necessary for strength-based sports [120,163]. This mesomorphic dominance is more pronounced in higher weight categories, where endomorphy also increases, with some individuals exceeding the limits of the somatoplot [31,164,165]. In contrast, other studies show that elite male tennis players often have an mesomorphic ectomorph somatotype, reflecting a lean and muscular build advantageous for the physical demands of tennis, contributing to agility, strength, and on-court performance [63,166]. Elite male surfers are characterized by high levels of strength, power, speed, and endurance, essential for explosive movements, sustained paddling, and the overall physical robustness required in surfing [86,167,168]. Although limited data are available on the body composition of pelotaris [91], it is suggested that their physical characteristics may resemble those of the previously mentioned athletes, which would be beneficial for the intense and dynamic demands of Basque pelota. This area of study could greatly benefit from additional research exploring the specific physical requirements of this sport.

Female athletes in individual sports exhibited a central somatotype (Appendix A), including gymnasts, sprint paddle athletes, and dancers specializing in ten dance and Latin dance. Latin dancers face greater physiological demands compared to ten dance performers, requiring significant aerobic capacity and high intensity during competitions. High-quality dance performances are characterized by specific movement patterns, such as hip sway and asymmetrical limb movements, which are crucial for effective execution and aesthetic appeal [169,170]. Gymnasts are typically characterized by small stature, low body mass, and a low body fat percentage. Compared to this study, other research has shown gymnasts to exhibit a mesomorphic ectomorph somatotype, with these traits being consistent across different competition levels and age categories. However, non-elite gymnasts may present slightly higher skinfold measurements [171,172,173]. This information is essential for coaches in athlete selection and training processes. Sprint paddle athletes display a lean body composition, well-developed upper body musculature, moderate height and body mass, and high aerobic capacity. Over time, there has been a trend toward a more compact and robust physique among sprint paddle athletes [174,175]. In individual sports, specific somatotypes and physical characteristics are observed, reflecting a tendency toward specialized body compositions that meet the physiological demands of each discipline.

The findings of this review have significant implications for sports science and the training of elite athletes. Sports specialization has driven distinct somatotype preferences, such as endomorphic elite kayakers, endo-mesomorphic basketball players, and ectomorphic football players, allowing athletes to meet their sport’s physical demands [176]. In endurance events like Ironman competitions, lower endomorphy and higher ectomorphy correlate with better performance, with athletes closer to a 1.7-4.9-2.8 somatotype excelling [32]. Advances in training methods show that targeted programs, such as those improving VO_2_max in middle-distance runners, can effectively alter somatotypes for better performance [12]. Regular training enhances motor skills and physiological fitness, especially during growth phases in childhood and adolescence [177]. These findings suggest that distinct somatotype patterns—such as the endomorphic mesomorph profile in many male athletes and the central classification in many female athletes—may influence injury risk due to varying biomechanical and physiological demands across sports. The higher body mass associated with endomorphic mesomorph athletes can increase joint stress during explosive movements. In contrast, the relatively lean physique of ectomorphic mesomorph athletes may elevate susceptibility to muscle strains and stress fractures under high-intensity or high-volume training. Moreover, central somatotypes, though balanced, still require careful management of conditioning and technique to avoid overuse injuries.

By determining that the endomorphic mesomorph somatotype predominates among male elite athletes and the central somatotype among female elite athletes, the substantial influence of anthropometric characteristics on sports performance is evident. This information is valuable for coaches, sports nutritionists, physical trainers, and healthcare professionals who design training programs and talent identification strategies. By aligning athletes with sports that suit their body types, they enhance performance and facilitate long-term athletic success. Training programs can be tailored to individual needs, while nutritional guidelines and recovery strategies are optimized to maximize each athlete’s innate potential and overall performance.

### 4.2. Limitations

Despite efforts to conduct a comprehensive search, our review has certain limitations that should be considered when interpreting the results. Although 66 studies on 43 sports disciplines were included, with an initial search of 155 studies, it is not possible to fully capture the diversity of somatotypes across all sports and levels of competition. In the case of female athletes, some disciplines may be underrepresented (e.g., soccer, basketball, paddling) or not represented at all (e.g., windsurfing, jiu-jitsu, boxing). Most studies in the literature focus on team and combat sports, which could influence the determination of the predominant somatotype. Additionally, cultural, genetic, and environmental differences, which also impact somatotype, were not accounted for [178]. Other inherent factors, such as the use of different calipers to assess body composition, present limitations due to variations in pressure, calibration, and the specific constraints of each device [179,180]. Since this scoping review included athletes from different countries, there are apparent differences in somatotypes among ethnic groups influenced by both genetic and environmental factors. These differences can impact athletic performance and are important considerations in sports science. While genetic predispositions play a role, environmental factors and lifestyle choices also significantly shape somatotype characteristics.

## 5. Conclusions

In this scoping review, it was determined that the predominant somatotype among male elite athletes was endomorphic mesomorph (32.8%), followed by balanced mesomorph (25.2%) and ectomorphic mesomorph (18.3%). Among female elite athletes, the predominant somatotype was central (31.5%), followed by endomorphic mesomorph (22.2%) and mesomorph–endomorph (20.4%). The primary somatotype of male elite athletes was characterized by high muscularity and a higher percentage of body fat, which was prevalent across various sports disciplines. However, variations were observed based on sport classification. For female elite athletes, the central classification showed an equal distribution among the relative components of the somatotype. In team sports, male athletes predominantly exhibited an endomorphic mesomorph classification, while female athletes showed an mesomorphic endomorph somatotype. In combat sports, athletes of both sexes shared the same endomorphic mesomorph classification. In speed and endurance sports, male athletes were predominantly ectomorphic mesomorph, while female athletes exhibited a central classification. Finally, in individual sports, male athletes showed an endomorphic mesomorph classification, whereas female athletes predominantly displayed a central classification.

Somatotype is highly relevant in the context of elite athletes, as it provides insights into the optimal body composition and physique for different sports, contributing to talent identification, personalized training, and performance enhancement. Understanding the somatotypical profiles of elite athletes enables coaches and sports scientists to tailor training programs and optimize selection processes more effectively. Assessing body physique is essential not only in sports generally but also based on player position, specialty, and category. The collected information and reference tables are helpful not only for elite athletes but also for those engaging in recreational exercise or beginning preparation for a specific sport.

Future research should prioritize increasing the representation of elite female athletes across diverse sports and competitive levels to enhance the understanding of somatotype-specific performance. Studies should examine how somatotype varies with hormonal fluctuations, age-related changes, and injury susceptibility, utilizing advanced technologies to provide deeper insights into female athlete profiles. Additionally, longitudinal studies are suggested to analyze how training programs may influence and modify somatotypes over time. Such studies would provide valuable insights into the dynamic nature of somatotypes and their adaptability to targeted interventions, offering a deeper understanding of the relationship between somatotypes, training adaptations, and performance outcomes in various sports disciplines.

## Figures and Tables

**Figure 1 sports-13-00038-f001:**
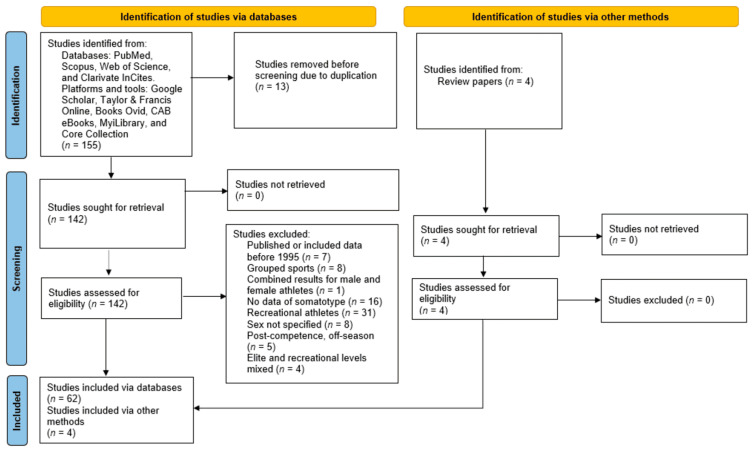
Flow diagram of the Preferred Reporting Items for Systematic Reviews and Meta-analyses extension for Scoping Reviews (PRISMA-ScR) [54].

**Figure 2 sports-13-00038-f002:**
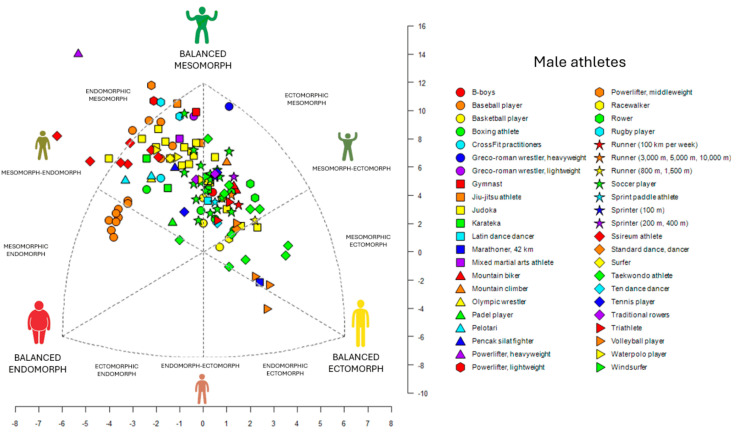
This chart showed the weighted mean somatotype (endomorphy, mesomorphy, ectomorphy) of 2322 male athletes across various sports. Each point represents the mean physique classification for a specific sport.

**Figure 3 sports-13-00038-f003:**
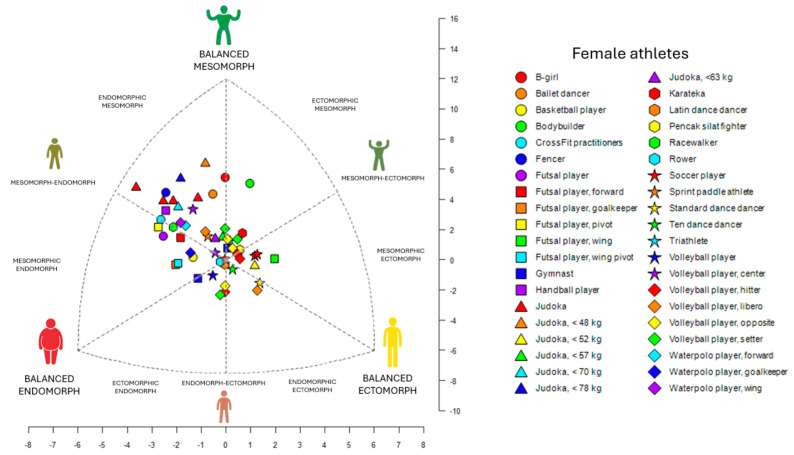
This chart shows the weighted mean somatotype (endomorphy, mesomorphy, ectomorphy) of 1435 elite female athletes across various sports. Each point represents the mean physique classification for a specific sport.

## Data Availability

The original contributions and data created presented in this study are included in the article/Appendix A. Further inquiries can be directed to the corresponding author.

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
