# Peer review of "The Shape of Success: A Scoping Review of Somatotype in Modern Elite Athletes Across Various Sports"

_sports, 2025, doi:10.3390/sports13020038_

Round 1
Reviewer 1 Report
Comments and Suggestions for Authors
The article "The Shape of Success: A Scoping Review of Somatotype in Modern Elite Athletes Across Various Sports" offers valuable insights for coaches and sports professionals focused on optimizing training and athlete selection. While the study is methodologically sound, and the results are presented clearly and comprehensively, its scientific contribution is somewhat limited. A more diverse range of sports could have enhanced the scientific impact. Additionally, comparing athletes' somatotypes with their athletic success/achievement could provide further insight into the relationship between body composition and performance.
Author Response
1. Summary |
|
|
Thank you very much for your thorough review and thoughtful comments. Your feedback has been invaluable in improving the quality and clarity of our manuscript. We greatly appreciate the time and effort you have dedicated to this review process.
|
||
2. Questions for General Evaluation |
Reviewer’s Evaluation |
Response and Revisions |
Is the work a significant contribution to the field? |
2/5 |
Thank you for your valuable feedback; it has greatly improved clarity and quality. |
Is the work well organized and comprehensively described? |
5/5 |
Thank you for your valuable feedback; it has greatly improved clarity and quality. |
Is the work scientifically sound and not misleading? |
1/5 |
Thank you for your valuable feedback; it has greatly improved clarity and quality. |
Are there appropriate and adequate references to related and previous work? |
4/5 |
Thank you for your valuable feedback; it has greatly improved clarity and quality. |
Is the English used correct and readable?
|
3/5 |
Thank you for your valuable feedback; it has greatly improved clarity and quality. |
3. Point-by-point response to Comments and Suggestions for Authors |
|
|
Comment 1: “The article "The Shape of Success: A Scoping Review of Somatotype in Modern Elite Athletes Across Various Sports" offers valuable insights for coaches and sports professionals focused on optimizing training and athlete selection. While the study is methodologically sound, and the results are presented clearly and comprehensively, its scientific contribution is somewhat limited. A more diverse range of sports could have enhanced the scientific impact. Additionally, comparing athletes' somatotypes with their athletic success/achievement could provide further insight into the relationship between body composition and performance.” Response 1: Thank you for your thoughtful comment and valuable suggestions. To address your feedback, we have created supplementary material that compares different elite levels of judo in modern elite athletes, providing additional insights into the relationship between somatotype and athletic performance. We appreciate your input, which has helped enhance the scientific contribution of our study. Correction 1: Figure S3 for supplementary material was created. Page 6, paragraph 4, lines 232-236 Given the presence of various elite levels, judokas were selected as the focus group due to their representation across a diverse spectrum of elite levels, including international competitors, senior athletes, South Korean national-level athletes, master belts (1st to 5th dan), black belts, and distinct weight categories. This analysis revealed differences in somatotypes, as illustrated in Figure S3 of the Supplementary Material. Page 10, paragraph 3, lines 384-298 The observed differences in somatotypes among judokas (Figure S3) at various elite levels provide valuable insights into the physical demands and morphological adaptations required for success in judo. Elite-level judokas, including international competitors and senior athletes, often exhibit somatotypes that favor strength and power, such as a higher balanced mesomorphic profile, due to the intensive physical requirements of the sport. In contrast, South Korean national-level judokas and master belts (1st to 5th dan) displayed differences despite having the same weight category classifications, demonstrating variations not only by elite level but also by ethnicity. This reflects a combination of strength, endurance, and technical skill development, which varies across weight categories. Heavier weight categories may be associated with higher endomorphic components, whereas lighter categories favor ectomorphic or mesomorphic profiles. These findings underscore the importance of somato-type-specific training and nutrition programs to optimize performance at different competitive levels and weight classes. The inclusion of judokas with diverse elite levels in this analysis broadens our understanding of how body composition and somatotype influence performance outcomes in judo.
|
Reviewer 2 Report
Comments and Suggestions for Authors
Thank you for the opportunity to review this article. The paper addresses a novel under-researched area, which has the potential to provide useful recommendations for coaches and physical trainers. However, there are some questions that need to be addressed to the manuscript.
ABSTRACT
- It is noted that the title includes a term already listed as a keyword. To avoid redundancy, it is suggested to replace this term in the keywords with another.
INTRODUCTION
- - Injury prevention is not addressed (lines 66-67).
- - Historical examples could be added (lines 67-69).
- - The impact of technological developments could be added (lines 61-62).
- Why do you consider this to be a problem? It is reasonable for somatotypes to be studied by discipline, as each has its own characteristic profile and specific demands. It would be helpful if the authors could clarify why previous research, focused on specific disciplines or conducted before 1995, might be insufficient or limiting for the purpose of this study (line 71-72)
- Change “study” to “scoping review” (line 79)
- Add in various sports: team sports, combat sports, speed and endutance sports and individual sports. (line 80)
MATERIAL AND METHODS
- Why was he not registered in PROSPERO? (line 86)
- The impact of factors related to ethnicity (such as individuals of Caucasian, Afro-descendant, Asian, Hispanic, or Indigenous origin, among others) is not addressed in depth. It is recommended to explore in greater detail how these variations might influence the study’s results and their interpretation (line 90)
- Specify which other languages (line 94)
- Why is the criterion of ‘non-professional athletes with at least 10 years of experience’ used? Is there any reference or evidence supporting the idea that athletes with over 10 years of experience can be classified as elite? It would be helpful to clarify or provide justification for this definition. (line 100)
- It would be helpful to clarify and differentiate which resources are structured databases (e.g., PubMed, Scopus, Web of Science) and which are platforms or tools (e.g., Google Scholar or Taylor & Francis Online) (line 115)
- Were all studies analysed between 1995 and 2024 or only a subset of them (April-September 2024)? Clarify it (line 117-118)
- To improve the bibliographic search strategy and ensure a comprehensive review, it is essential to consider the use of truncation, related additional terms, and Boolean operators. Truncation is a key tool that enables the inclusion of all grammatical variations and orthographic forms of a term, significantly broadening the scope of the search. For instance, using “athlet*” will capture words such as athlete, athletes, athletic, or athletics, covering different contexts in which the term may be used. Similarly, “anthropometr*” will capture terms like anthropometric, anthropometry, and anthropometrical, which are common in studies analysing physical or bodily measurements. Truncations such as “composit*”, “morphol*”, or “somatotyp*” are also useful for encompassing key concepts such as body composition, morphology, and somatotype classification without restricting the search to a single term.
- Moreover, it is important to include additional terms that broaden the thematic focus of the search. For “body composition”, terms like body fat percentage, lean body mass, adiposity, or specific techniques such as DEXA or bioelectrical impedancecan be added, as these methodologies are frequently used in related studies. For “anthropometric”, terms such as waist-to-hip ratio, circumference measurements, or growth patterns could enrich the search, particularly when linking physical measurements to athletic performance. Regarding “somatotype”, incorporating synonyms or related terms such as physique classification, biotype, or specific categories (endomorphic, mesomorphic, ectomorphic) may be key to capturing studies that employ alternative approaches to describing body structure.
- When addressing “elite athlete” or “athlete”, it is important to include terms such as professional player, high-performance athlete, Olympic athlete, or even sub-elite athlete, as many studies use different ways to describe high-level athletes. This helps include relevant studies that might otherwise be excluded in an overly narrow search. It is also beneficial to consider synonyms such as sportsman, sportswoman, or sportsperson, depending on the cultural context and language in which the articles are written.
- Strategic use of Boolean operators is also critical. Combinations like “athlete” OR “sportsman” OR “sportswoman”broaden the search by including synonyms, while “body composition” AND “elite athlete” ensures that the results encompass both key concepts. Additionally, the operator NOT can be useful for excluding irrelevant terms, such as in “athlete” NOT “recreational”, if the aim is to focus on competitive athletes. (line 123)
- Add year (line 135)
- - Add the number of subjects (n=X) in the description of the figure. (line 252 and 256)
DISCUSSION
- While practical implications are mentioned, could the authors provide concrete examples of how the results might be applied to talent identification or training programme planning?
- Could the discussion include an analysis of how the results may influence injury risk, providing further insights into their practical relevance?
- Could the authors propose longitudinal studies as a future research direction to analyse how training programmes may influence and modify somatotypes over time
Reviewer 3 Report
Comments and Suggestions for Authors
The introductory section is clear and well-structured. The authors effectively outline the theoretical context and justify the importance of somatotype testing in the context of competitive sports. They refer to classical and contemporary sources, emphasizing the development of somatotype assessment methodology over the years.
Although the purpose of the study was clearly stated, the introduction of a working hypothesis or a specific research question could have further focused the reader's attention and strengthened the scientific value of the article.
Suggestion: Add a fragment such as:
"This study asks the question: What are the contemporary patterns of elite athlete somatotypes in different sports?"
The authors briefly mention somatotype differences but do not develop this issue at the introductory level. Adding a few examples could have better prepared the ground for the results of the analysis.
Although the results section includes differences in the somatotypes of men and women, the introduction omits this aspect. Providing such context at the beginning could have enriched the narrative and presented a more complete picture of the issue.
The article could additionally take into account the influence of genetics and environment on the formation of somatotype, which would enrich the theoretical basis of the study.
The Materials and Methods section is extensive and precise, in line with international standards for literature reviews. The authors use PRISMA-ScR protocols and Joanna Briggs Institute (JBI) guidelines, which demonstrates methodological integrity and attention to transparency.
The inclusion and exclusion criteria, the selection process, and the method of data extraction are described in detail, which increases the credibility of the review.
Despite the detailed description of the selection process, no mention was made of the quality assessment of the included articles (e.g. AMSTAR or ROBIS tools). A quality assessment of the studies could have highlighted the robustness of the results and excluded potentially weak sources.
The section states that the data were “reviewed by the authors,” but there is no detailed description of the review process. What were the criteria for resolving discrepancies? Was interrater agreement (e.g., kappa) used?
The authors selected studies from 1995 onwards, arguing that athletes have undergone significant morphological changes since then. However, the lack of detailed justification for this time limit may raise questions.
The Results section provides extensive and detailed data on the somatotypes of elite athletes from various sports. The results are presented in both text and graphical form (somatotype charts on page 7).
The authors clearly present the main somatotype categories, gender differences, and specific somatotype characteristics by sport (team, individual, endurance, and combat). The data are presented in a clear and logical manner, making them easy to interpret.
The Discussion section is extensive and relates well to the results presented in the Results section. The authors interpret the data in the context of existing literature and point out the practical implications of the results, which underscores the importance of the study. The discussion includes comparisons between different disciplines and genders, pointing out differences in athlete somatotypes and their implications for sports performance and talent selection.
The article does not take into account the influence of geographical and cultural differences on somatotypes. It is known that athletes from different regions of the world may differ in terms of body composition due to genetic and environmental differences.
In the discussion, the authors analyze somatotypes in terms of team, individual, combat, and endurance sports, but there is no detailed analysis within these categories. For example, combat sports include various disciplines (judo, MMA, boxing) that may differ in physical demands.
Although the authors mention the need for greater representation of women and diverse disciplines, there are no detailed recommendations for future research directions.
Round 2
Reviewer 2 Report
Comments and Suggestions for Authors
The authors have taken my contributions into consideration in order to improve the level of the manuscript, which is very interesting.